# Detection of Human Papillomavirus DNA in Paired Peripheral Blood and Cervix Samples in Patients with Cervical Lesions and Healthy Individuals

**DOI:** 10.3390/jcm10215209

**Published:** 2021-11-08

**Authors:** Kamylla Conceição Gomes Nascimento, Élyda Gonçalves Lima, Zhilbelly Mota Nunes, Marconi Rêgo Barros Júnior, Marcus Vinícius de Aragão Batista, Antonio Roberto Lucena Araujo, Jacinto da Costa Silva Neto, Bárbara Simas Chagas, Ana Pavla Almeida Diniz Gurgel, Antonio Carlos de Freitas

**Affiliations:** 1Laboratory of Molecular Studies and Experimental Therapy (LEMTE), Department of Genetics, Federal University of Pernambuco, Recife 50670-901, Brazil; kamylla.conceicao@ufpe.br (K.C.G.N.); elyda.lima@gmail.com (É.G.L.); marconijrr@gmail.com (M.R.B.J.); babisimas@gmail.com (B.S.C.); antonio.cfreitas@ufpe.br (A.C.d.F.); 2Laboratory of Molecular Genetics (LAGEM), Department of Engineering and Environment, Federal University of Paraiba, João Pessoa 58297-000, Brazil; zhilbelly@hotmail.com; 3Laboratory of Molecular Genetics and Biotechnology (GMBio), Department of Biology, Federal University of Sergipe, São Cristóvão 49100-000, Brazil; genetics_marcus@hotmail.com; 4Department of Biophysics and Radiobiology, Federal University of Pernambuco, Recife 50670-901, Brazil; araujoarl@gmail.com; 5Department of Histology, Federal University of Pernambuco, Recife 50670-901, Brazil; jacinto.costa@ufpe.br

**Keywords:** human papillomavirus DNA, cervix, peripheral blood, cervical lesions, healthy individuals

## Abstract

This study evaluated the presence of Human Papillomavirus (HPV) DNA in the cervix and peripheral blood of women with cervical intraepithelial neoplasia (CIN I, II, and III) and healthy individuals. Overall, 139 paired peripheral blood and cervix samples of healthy women and women with CIN I, II, and III (*n* = 68) were tested for HPV DNA by using standard procedures. Polymerase chain reaction (PCR) sequencing determined HPV types. Quantification of HPV16 E6 and E2 genes was performed to determine viral load and physical state. HPV DNA was detected in the cervix (21.1% in healthy individuals; 48.8–55.5% in CIN patients), blood (46.4% in healthy individuals; 44.1–77.7% in CIN patients) and paired peripheral blood and cervix samples (24% in healthy individuals; 32.5–44.4% in CIN patients). The most frequent types found in the cervix were HPV16, 18, 31, 33, 58, and 70, while HPV16, 18, 33, 58, and 66 were the most frequent types found in the blood. HPV DNA in the cervix was associated with previous sexually transmitted infections (STIs) (*p* = 0.023; OR: 2.978; CI:1.34–7.821), HPV DNA in the blood (*p* = 0.000; OR: 8.283; CI:3.700–18.540), and cervical lesions (CIN I/II or III) (*p* = 0.007). Binomial logistic regression showed that HPV DNA in the blood (*p* = 0.000; OR: 9.324; CI:3.612–24.072) and cervical lesions (*p* = 0.011; OR: 3.622; CI:1.338–9.806) were associated with HPV DNA in the cervix. However, we did not find an association between HPV DNA in the blood and cervical lesions (*p* = 0.385). Our results showed that only HPV DNA found in the cervix was associated with cervical lesions.

## 1. Introduction

Cervical cancer is the fourth most common malignancy diagnosed in women worldwide, accounting for 6.6% of total cancer cases [1]. Nearly 90% of deaths caused by cervical cancer occur in women from middle- and low-income countries [1].

It is well established that persistent infection caused by Human Papillomaviruses (HPV) is involved in cervical and head and neck carcinogenesis [2,3,4,5,6,7,8,9]. Despite the advances in vaccination, treatment, and monitoring, cervical cancer is still a significant cause of death in middle- and low-income developing countries [1]. Hence, there is a need for minimally invasive biomarkers to monitor cervical cancer and cervical lesions. Recent studies have shown that circulating free DNA (cfDNA) or circulating tumor DNA (ctDNA) may be valuable tools for monitoring cancers caused by HPV [10,11,12,13,14,15,16]. In this scenario, several studies have demonstrated the presence of HPV DNA in peripheral blood mononuclear cells [17,18,19,20,21,22,23,24,25], serum [26,27], and plasma [28,29,30] from cervical cancer and non-cervical cancer patients. However, despite its importance, few studies have evaluated the HPV DNA in paired peripheral blood and HPV DNA in the cervix from precancerous cervical patients and healthy individuals [21,22,23,24,30,31,32]. Thus, this study evaluated the prevalence of HPV DNA and viral load in paired peripheral blood and cervix samples in cervical intraepithelial neoplasia (CIN I, II, and III) patients and healthy individuals.

## 2. Materials and Methods

### 2.1. Patients

The paired cervix and blood samples were obtained from women during their gynecological consultations at Hospital das Clínicas, Pernambuco State, Northeast Brazil. The samples were collected from November 2010 to December 2016. A total of 139 women agreed and signed free and informed consent forms. This study was approved by the Ethics and Research Committee (CAAE: 0058.0.106.000-10, HUOC/PROCAPE 64/2010) and (CEP/CCS/UFPE under CAAE: 09307612.8.0000.5208). In this study, we examined 139 paired cervix and blood samples. Among these, 71 (51%) were from healthy individuals, 43 (31%) patients had cervical intraepithelial cervical grade I (CIN I), 16 (11.5%) patients had cervical intraepithelial cervical grade II (CIN II), and 9 (6.5%) patients had cervical intraepithelial cervical grade III (CIN III). Cervical cancer samples (paired cervix and blood samples) were excluded due to possible metastasis processes.

### 2.2. HPV DNA Cervix and Blood Samples

DNA from cervical and peripheral blood was extracted using the DNeasy Blood & Tissue Kit (QIAGEN GmbH, Hilden, Germany), following the manufacturer’s instructions. Subsequently, all samples were quantified using a spectrophotometer (NanoVue Plus Spectrophotometer). DNA quality was confirmed by polymerase chain reaction (PCR), which amplifies a fragment of the β-globin gene with the primers described in Table 1.

The presence of HPV DNA was assessed by amplification with two sets of oligonucleotides. The first step was performed with the set of consensus and degenerate primers MY09/11 (Table 1), which anneal in a conserved region of the L1 gene, amplifying a product of approximately 450 bp. Subsequently, nested PCR was performed using oligonucleotides GP05/06, generating a fragment of approximately 140 bp. The DNA extracted from CaSki cell lineages was used as a positive control in all PCR reactions. Besides, all tests were performed in triplicate. 

HPV-positive samples were genotyped by PCR followed by sequencing. All the HPV positive samples were sequenced by the dideoxy-terminal fluorescent method by using the ABI PRISM BigDyeTM Terminator Cycle Sequencing v 3.1 Ready Reaction kit (Applied Biosystems, Waltham, MA, USA). The sequences were analyzed using the Staden package (STADEN, 1996) with the programs Gap4 (version 4.0) and Pregap4 (version 1.5), which were used to assemble the contigs of the sequences obtained from HPV DNA. After the contig assembly, the BLAST program, available at http://blast.ncbi.nlm.nih.gov/Blast.cgi (accessed on 31 December 2016), was used to compare previously known HPV sequences.

### 2.3. Viral Load and Physical Status

Real-time PCR (qPCR) was used to determine the viral load and physical status of HPV16-positive patients. For this, seven random paired cervix and blood samples derived from CIN I, II, and III patients and 12 samples derived from patients with normal cervical cytology were selected. HPV16 viral load was determined through E6 gene levels present in all patient groups. Each sample’s viral load was expressed as the number of copies of E6 in 50 ng of DNA [33]. A series of dilutions of the complete HPV genome cloned into the pBR-322 vector was carried out to generate the standard curve. Calibration curves for the E2 and E6 genes were created by using the standard dilutions. The DNA extracted from CaSki and SiHa cell lineages was used as a positive control for the reactions. Once calibration curves were constructed for each gene, the samples were analyzed in duplicate. As for physical status, the virus was considered integrated into the host genome when the E2 gene was not detected. E2/E6 ratio was calculated to differentiate the episomal from the mixed physical status. E2/E6 ratio <1 indicates the mixed physical status; values >1 indicate the predominance of the episomal physical status [33].

### 2.4. Statistical Analysis

Clinical aspects of patients were compared between patients with HPV DNA-positive and HPV DNA-negative samples from the cervix by using chi-square and Fisher’s exact tests. We performed hierarchical binary logistic regression analysis to identify the variables associated with HPV positive in the cervix, HPV DNA in blood, and cervical lesions. The correlation of viral load of HPV16 DNA in paired cervical and blood samples was examined using the Spearman’s rho and Kendall´s Tau_b correlation coefficient tests. All *p*-values were two sided with a significance level of 0.05, except the Spearman´s rho and Kendall´s Tau_b correlations. All data were analyzed using SPSS version 26 (SPSS Inc., Chicago, IL, USA).

## 3. Results

### 3.1. Population

We performed a screening of HPV DNA in a total of 139 paired cervix and blood samples from healthy individuals and CIN patients. In total, HPV DNA was found in 49% (68/139) of blood samples and 37.4% (52/139) of cervical samples. Patients’ socio-demographic characteristics, such as age, ethnicity, age at first sexual relation, number of pregnancies, contraceptive use, number of sexual partners, smoking, previous STIs, and cervical lesions, were compared according to HPV DNA positive and negative results in cervical samples. HPV DNA in the cervix was associated with previous STIs (*p* = 0.023; OR: 2.978; CI:1.34–7.821), HPV DNA in blood (*p* = 0.000; OR: 3.369; CI:3.700–18.540), and cervical lesions (*p* = 0.007) (Table 2, Appendix A).

### 3.2. HPV DNA Detection in Cervix and Blood Samples

HPV DNA was detected in the cervix of 24% of healthy individuals, 48.8% of patients with CIN I, 56% with CIN II, and 55.5% with CIN III. In peripheral blood, HPV DNA was found in 46.4% of healthy individuals, 44.1% of patients with CIN I, 55.5% with CIN II, and 77.7% with CIN III. Regarding the cervix or peripheral blood samples, HPV DNA was detected in 46.4% of healthy individuals, 60% of patients with CIN I, 75% with CIN II, and 88.8% with CIN III. In paired cervix and peripheral blood samples, HPV DNA was found in 24% of healthy individuals, 32.5% of patients with CIN I, 38% with CIN II, and 44% with CIN III (Figure 1, Appendix A).

The most frequently found genotypes in the cervix were HPV16, 18, 31, 33, 58, and 70, while HPV16, 18, 33, 58, and 66 were the most frequently found blood sample types. Globally, the genotypes most frequently found were HPV16 and HPV58 in both cervix and peripheral blood samples. Additionally, HPV16 was the most frequent genotype in the paired cervix and blood samples (Figure 2).

Binomial logistic regression showed that HPV DNA in the blood (*p* = 0.000; OR: 9.324; CI:3.612–24.072) and cervical lesions (*p* = 0.011; OR: 3.622; CI:1.338–9.806) were associated with HPV DNA in the cervix. In contrast, only HPV DNA in the cervix (*p* = 0.011; OR: 3.918; CI:1.366–11.239) was associated with HPV DNA in the blood (*p* = 0.00). In addition, we did not observe an association between HPV DNA in blood and cervical lesions (*p* = 0.385).

### 3.3. Evaluation of Viral Load in Paired Cervix/Blood Tissues

Since HPV16 is the most frequent genotype found in both blood and the cervix tissues, we evaluated the viral load of HPV16 in the paired cervix and blood samples. Cervix and peripheral blood samples from 12 patients with CIN I, II, and III, as well as from 17 healthy individuals were selected for viral load quantification. Regarding the CIN patients, the viral load varied from 3.65 to 21.58 (E6 copies/50 ng of DNA) in blood and 0.2 to 138 (E6 copies/50 ng of DNA) in the cervix. A correlation was observed between HPV16 viral load present in the cervix and peripheral blood samples in patients with CIN II/III (r = 0.65) and CIN III (r = 0.60) Appendix A).

We did not observe a correlation between HPV16 viral load in the cervix and peripheral blood samples in patients with CIN II/III (r = 0.25) and CIN III (r = 0.20) (Appendix A). Additionally, there was no correlation between HPV DNA viral load in the cervix and blood in samples from women with the integrated or mixed virus in the cervix (r = 0.0) or mixed virus in their peripheral blood (r = −0.11) (Appendix A).

Regarding the healthy individuals, the viral load varied from 2.76 to 83.89 (E6 copies/50 ng of DNA) in blood and from 0.12 to 246 (E6 copies/50 ng of DNA) in the cervix. The integrated, episomal, and mixed physical status were observed in peripheral blood and cervix samples. However, we did not observe a correlation between HPV DNA viral load in the cervix and peripheral blood samples in healthy patients (r = −0.15).

## 4. Discussion

This study investigated the presence of HPV DNA in paired peripheral blood and cervix samples from women with cervical lesions and from healthy individuals. Our results showed HPV DNA in the cervix and peripheral blood samples of both healthy individuals and CIN patients. In addition, we also observed a significant association between the HPV DNA in the cervix and cervical lesions. However, we did not observe an association between HPV DNA in blood and cervical lesions. Moreover, we did not find a positive correlation between HPV16 viral load in the paired cervix and blood samples in women according to CIN grade and physical status of the virus in the cervix and blood. Globally, our results showed the presence of HPV DNA in the cervix and peripheral blood from cervical lesions patients and healthy individuals. However, only HPV DNA found in the cervix was associated with cervical lesions.

Several studies have demonstrated the presence of HPV DNA in blood from cervical cancer [17,18,19,27,28,29,31,34,35,36], cervical lesions [30], breast cancer [26], lung cancer [20], and head and neck cancer [37] patients. In addition, HPV DNA in blood was found in healthy individuals [21,22,23,24]. According to these studies, the prevalence of HPV DNA in blood ranged from 6.9% to 87%. The studied population, the type of lesion, the methodology used to detect HPV DNA, and the specimen type (serum, plasma, and peripheral blood) may reflect the HPV DNA prevalence variation in peripheral blood. Our study found a high prevalence of HPV DNA in the blood of patients with cervical lesions from Northeastern Brazil, where diseases caused by HPV are a significant public health concern [38]. HPV16 and HPV58 were the most frequent genotypes found in the cervix and blood of patients. We used PCR sequencing to detect HPV DNA in whole peripheral blood and HPV DNA in the cervix. Although PCR sequencing failed to detect multiple HPV types, this methodology is suitable for HPV DNA detection.

In this study, we also observed a direct relationship between the increase in prevalence of HPV DNA in blood and the severity of the lesions in patients with CIN I (44.1%), II (50%), and III (77.7%). Similar results were observed in previous studies regarding patients with cervical lesions [30]. However, we did not observe a positive correlation between the HPV16 viral load in the paired cervix and blood samples, according to the severity of the lesions (r = 0.25, CIN II/III; R = 0.20, CIN III). These findings follow a similar previous study, in which the authors did not find an association between viral load in the paired cervix and blood and CIN [30]. However, these results should be interpreted with caution since we performed a viral load analysis with few patients. In contrast, a positive correlation between HPV DNA in blood with the severity or tumor size was observed in cervical [11,17,27,34,35,36], head and neck [39,40], lung [20], and breast cancer [41]. Thus, more studies are needed to clarify whether HPV DNA viral load can be used to monitor cervical lesions.

To date, few studies discuss the presence of HPV DNA in the blood of healthy subjects [22,24,32]. HPV DNA has also been detected in pediatric patients [22,42], healthy blood donors [22,23,32], and normal smears [25,30]. Here, we observed HPV DNA in 24% of the paired cervix and blood samples from non-cervical lesion patients. When put together, these findings confirm the presence of HPV in blood from subjects with no cervical HPV infection history. However, it is impossible to exclude the possibility that the high HPV DNA prevalence in healthy individuals is due to HPV infection in other body sites. 

One of the HPV-induced process’s key events is integrating the virus’s genome into the host’s genome [43]. The state of integration is fundamental for the initiation of viral replication. One clue that HPV DNA is integrated into the host genome is the loss of E2. Therefore, absolute quantification of the E2 gene was performed, aiming to determine the viral genome’s physical status in peripheral blood samples from patients with cervical lesions and healthy individuals. Both groups presented predominance of the mixed form. However, previous studies have reported a predominance for integrated [35] and episomal forms [22]. The virus’s presence in mixed form and a high viral load in healthy individuals reinforces the hypothesis that the blood cells carry the virus. Further studies are needed to clarify this aspect.

HPV DNA in peripheral blood has been historically detected in cervical cancer and non-cervical cancer tumors, in which ctHPVDNA migrates from local infection to blood [22]. Furthermore, recent studies have shown cfHPVDNA in the blood of cervical cancer patients [10,11,27]. HPV DNA in the blood of patients with cancer can be originated through apoptosis, necrosis, nucleosomes, phagocytosis, extracellular vesicles, HPV carrier cells, and a combination of these mechanisms [44]. However, the dissemination of HPV DNA from cervical lesions and a resulting normal smear still needs to be understood. Our study found HPV DNA in peripheral blood from both patients with and without cervical lesions. In addition, we did not observe an association between HPV DNA in blood and cervical lesions. However, we found an association between HPV DNA in the cervix and cervical lesions. HPV DNA in blood could result from infections in other body sites, such as the head and neck. Hence, we speculated that the virus causes lesions that induce immune responses, predominantly lymphocytic, and then the cells would carry HPV DNA through the hematogenous route. In this scenario, studies focusing on Bovine papillomavirus (BPV)-infected cattle, a model for HPV infection study, support this hypothesis. These studies demonstrated BPV DNA in cattle’s peripheral blood, with or without warts [45,46,47,48,49,50,51] and viral activity in lymphocytes [52]. Previous studies have suggested that the blood can be a viral reservoir for BPV [45,50,52]. Studies in humans also support this hypothesis. For instance, some works have demonstrated HPV DNA in lymphocytes, dendritic cells, NK cells, neutrophils, B lymphocytes CD20^+^, and CD56^+^ [23,24].

In conclusion, we detected HPV DNA in the cervix and peripheral blood from patients with cervical lesions and from healthy individuals. However, only HPV DNA found in the cervix was associated with cervical lesions. More studies are needed to clarify the role of HPV DNA in blood and tumorigenesis.

## Figures and Tables

**Figure 1 jcm-10-05209-f001:**
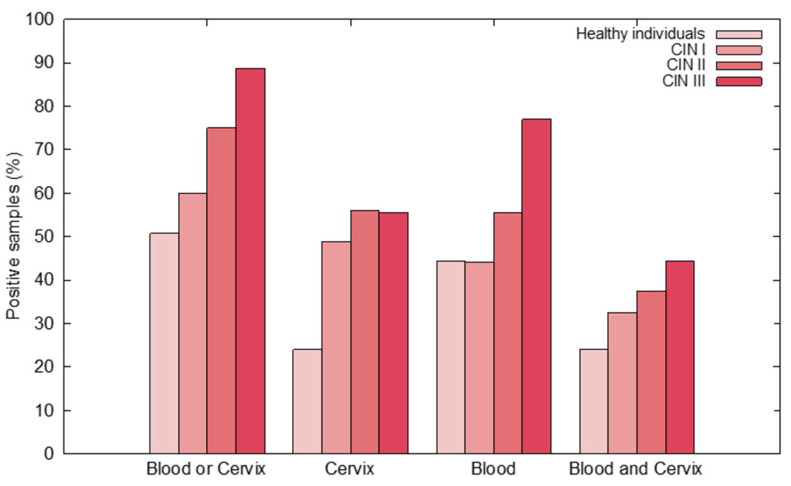
Prevalence of HPV in healthy individuals and CIN patients.

**Figure 2 jcm-10-05209-f002:**
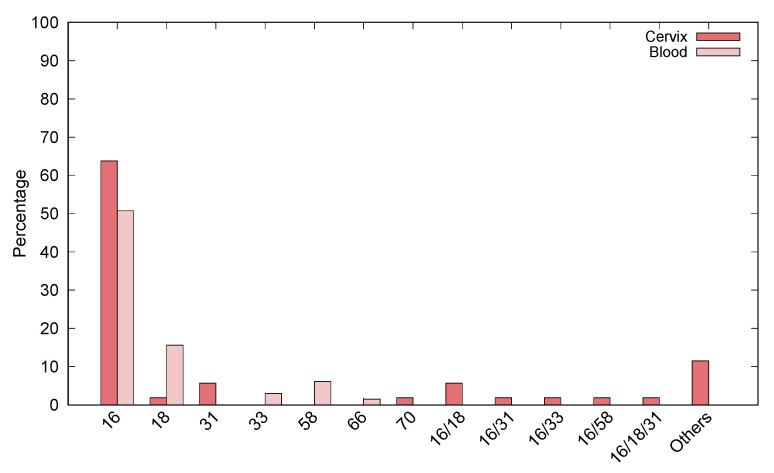
Frequency of HPV types found in the cervix and peripheral blood from women of Northeastern Brazil.

**Table 1 jcm-10-05209-t001:** Primers used for PCR amplification, sequencing, and viral load analysis.

	Sequence (5′3′)	Size (bp)
*β*-globina		110 bp
PC04 GH20	ACACAACTGTGTTCACTAGC CAACTTCATCCACGTTCACC	
MY09 MY11	CGTCCMARRGGAWACTGATC GCMCAGGGWCATAAYAATGG	450 bp
GP5 GP6	TTTGTTACTGTGGTAGATAC GAAAAATAAACTGTAAATCA	110 bp
E6 HPV16		
Forward	GAGAAACTGCAATGTTTCAGGACC	81 bp
Reverse	TGTATAGTTGTTTGCAGCTCTGTGC	
E2 HPV16		
Forward	AACGAAGTATCCTCTCCTGAAATTATTAG	76 bp
Reverse	CCAAGGCGACGGCTTTG	

MY09/11 degenerated primers: M = A or C, W = A or T, Y = C or T, and R = A or G.

**Table 2 jcm-10-05209-t002:** Clinical and histological characteristics of patients according to HPV status in the cervix.

		HPV Negative (Cervix) N° (%)	HPV Positive (Cervix) N° (%)	*p*-Value	OR (IC)
**Age**	<45	53 (38.1%)	31 (22.3%)	0.937	
	46–60	32 (23.0%)	20 (14.4%)		
	>60	2 (1.4%)	1 (0.7%)		
	Total	87 (62.6%)	52 (37.4%)		
**Residence**	Rural	13 (9.6%)	5 (3.7%)	0.382	
	Urban	72 (53.3%)	45 (33.3%)		
	Total	85 (63.0%)	50 (37.0%)		
**Ethnicity**	Black	15 (10.8%)	8 (5.8%)	0.791	
	Other	35 (25.2%)	24 (17.3%)		
	White	37 (26.6%)	20 (14.4%)		
	Total	87 (62.6%)	52 (37.4%)		
**Age at the first sexual relation**	<20	67 (48.2%)	44 (31.7%)	0.824	
	20–30	17 (12.2%)	7 (5.0%)		
	30–40	2 (1.4%)	1 (0.7%)		
	>40	1 (0.7%)	0 (0.0%)		
	Total	87 (62.6%)	52 (37.4%)		
**Number of pregnancies**	1	17 (16.2%)	8 (7.6%)	0.209	
	2–3	42 (40%)	15 (14.2%)		
	>4	11 (10.5%)	12 (11.4%)		
	Total	70 (66.7%)	35 (33.3%)		
**Contraceptive use**	No	70 (50.4%)	43 (30.9%)	0.744	
	Yes	17 (12.2%)	9 (6.5%)		
	Total	87 (62.6%)	52 (37.4%)		
**Smoking**	No	70 (50.4%)	42 (30.2%)	1.00	
	Yes	10 (7.2%)	6 (4.3%)		
	Ex-smoking	7 (5.0%)	4 (2.9%)		
	Total	87 (62.6%)	52 (37.4%)		
**Number of sexual partners**	1	38 (29.2%)	28 (21.5%)	0.779	
	2–3	33 (25.5%)	15 (11.5%)		
	>4	10 (7.7%)	6 (8.5%)		
	Total	81 (62.3%)	49 (37.7%)		
**Previous STIs**	No	67 (56.8%)	30 (25.4%)	0.023 *	2.978 (1.134–7.821)
	Yes	9 (7.6%)	12 (10.2%)		
	Total	76 (64.4%)	42 (35.6%)		
**HPV in blood**	No	60 (43.2%)	11 (7.9%)	0.000 *	8.283 (3.700–18.540)
	Yes	27 (19.4%)	41 (29.5%)		
	Total	87 (62.6%)	52 (37.4%)		
**CINI/II or III**	Without lesion	54 (38.8%)	17 (12.2%)	0.007 *	
	CIN I	22 (15.8%)	21 (15.1%)		
	CIN II	7 (5.0%)	9 (6.5%)		
	CIN III	4 (2.9%)	5 (3.6%)		
	Total	87 (62.6%)	52 (37.4%)		

Data are presented as the number of patients and percentage. * *p*-values were calculated by chi-square test or Fisher´s exact test. OR = Odds Ratio.

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
