# Peer review of "Detection of Human Papillomavirus DNA in Paired Peripheral Blood and Cervix Samples in Patients with Cervical Lesions and Healthy Individuals"

_jcm, 2021, doi:10.3390/jcm10215209_

Round 1
Reviewer 1 Report
Thank for the opportunity to review this manuscript on the detection of HPV DNA in paired peripheral blood and cervical samples across a spectrum on women with normal to severely dysplastic cervices. The manuscript reads well. The arguments presented are logical and well developed.
Pg 8 Can the authors explain why the PCR-sequencing failed to detect multiple HPV types.
Pg 8 Can the authors explain why there would be an association between HPV in blood in lung and breast cancer???
Typo Pg 8 "breast c\ancer"
I only had access to one supplementary table but the manuscript refers to 2 such tables.
Author Response
Reviewer # 1:
Thank for the opportunity to review this manuscript on the detection of HPV DNA in paired peripheral blood and cervical samples across a spectrum on women with normal to severely dysplastic cervices. The manuscript reads well. The arguments presented are logical and well developed.
Comment #1: Pg 8 Can the authors explain why the PCR-sequencing failed to detect multiple HPV types.
Response: We used Sanger sequencing. In this methodology, we performed a PCR using specific primers to sequencing a particular fragment. For this reason, we can not obtain several reads.
Comment #2: Pg 8 Can the authors explain why there would be an association between HPV in blood in lung and breast cancer???
Response: It weel-know that the transmission of HPV occurs through sexual intercourse. This infection can be eliminated or progress to cervical intraepithelial neoplasia (CIN) and cancer. Alternatively, HPV could infect lymphocytes and disseminate to other body sites, such as the breast and lung, through the hematological pathway.
Comment #3: Typo Pg 8 "breast c\ancer"
Response: We corrected this sentence.

Reviewer 2 Report
This manuscript is well written, however I find some shortcomings in the analysis, specifically in the evaluation of viral load in paired cervix/blood tissues.
First, I’m not sure what this sentence means: “We examined the viral load in the paired cervix and peripheral blood in HPV16 positive view that this genotype was most prevalent.” (lines 162-163)
I find Figure 4 very confusing. The axes are labelled identically, but the titles are different. The text offers little help. More troubling, it appears that a single patient (upper right) is responsible for whatever correlation the authors claim to have found in the 3 figures depicted, as well as the one not depicted (CIN II/III). Simple visual inspection of these figures confirms this.
Most troubling, there are no correlations between these data points. The inflated r-value is almost solely attributable to one outlier. If the authors want to fairly test the strength of their associations they should use a Spearman (monotonic) rather than a Pearson (linear) correlation. There are no linear associations here.
In lines 154-156, the authors refer to Table 3. I cannot find Table 3, and I can’t find the OR’s they note. Was Table 3 deleted? Are the binomial logistic regressions they cite bivariate or multivariate?
In Table 2, the p-values reported for the chi-square tests should be based on exact tests, rather than asymptotic tests. It is almost impossible to find a significant p-value for a chi-square when you have 1 or 2 counts in a cell.
Author Response
Reviewer # 2: This manuscript is well written, however I find some shortcomings in the analysis, specifically in the evaluation of viral load in paired cervix/blood tissues.
Comment #1: First, I'm not sure what this sentence means: "We examined the viral load in the paired cervix and peripheral blood in HPV16 positive view that this genotype was most prevalent." (lines 162-163).
Response: We replaced the above sentence by: "Since that HPV is the most frequent genotype found in both blood and the cervix tissues, we evaluated the viral load of HPV16 in the paired cervix and blood samples".
Comment #2: I find Figure 4 very confusing. The axes are labelled identically, but the titles are different. The text offers little help. Most troubling, there are no correlations between these data points. The inflated r-value is almost solely attributable to one outlier. If the authors want to fairly test the strength of their associations they should use a Spearman (monotonic) rather than a Pearson (linear) correlation. There are no linear associations here. More troubling, it appears that a single patient (upper right) is responsible for whatever correlation the authors claim to have found in the 3 figures depicted, as well as the one not depicted (CIN II/III). Simple visual inspection of these figures confirms this.
Response: We deleted Figures 3 and 4A/4B. We recalculate the correlation coefficient (r) using Spearmen instead Pearson´s tests because our N (number of patients) is below 30. By using Spearmen´test, we did not observe a correlation between the paired cervix and blood HPV16 viral load. Therefore, the modifications in the manuscripts were following (in bold):
1- Statistical analysis
Clinical aspects of patients were compared between patients with HPV DNA positive and negative in the cervix by using chi-square and Fisher's exact tests. We performed hierarchical binary logistic regression analysis to identify the variables associated with HPV positive in the cervix, HPV DNA in blood, and cervical lesions. The correlation of viral load of HPV16 DNA in paired cervical and blood samples was examined using the Spearman´s rho and Kendall´s Tau_b correlation coefficient tests. All P-values were two-sided with a significance level of 0.05, except the Spearman´s rho and Kendall´s Tau_b correlations. All data were analyzed using SPSS version 26 (SPSS Inc.Illinois, USA).
2- Results:
We did not observe a correlation between HPV16 viral load in the cervix and peripheral blood samples in patients with CIN II/III (r=0.25) and CIN III (r=0.20) (Supplementary data 2). Also, there was no correlation between HPV DNA viral load in the cervix and blood in samples from women with the integrated or mixed virus in the cervix (r=0.0) and in mixed in their peripheral blood (r=-011) (Supplementary data 1).
Regarding the healthy individuals, the viral load varied from 2.76 to 83.89 (E6 copies/50ng of DNA) in blood and 0.12 to 246 (E6 copies/50ng of DNA) in the cervix. The integrated, episomal, and mixed physical status were observed in peripheral blood and cervix samples. However, we did not observe a correlation between HPV DNA viral load in the cervix and peripheral blood samples in healthy patients (r=-0.15).
- Discussion:
In this study, we also observed a direct relationship between the increase of prevalence of HPV DNA in blood and the severity of the lesions in patients with CIN I (44.1%), II (50%), and III (77.7%). Similar results were observed in previous studies regarding patients with cervical lesions [28,30]. However, we did not observe a positive correlation between the HPV16 viral load in the paired cervix and blood samples, according to the severity of the lesions (r=0.25, CIN II/III; R=0.20, CIN III). These findings follow a similar previous study, in which they did not find an association between viral load in the paired cervix and blood and CIN [30]. However, these results should be interpreted with caution since we performed a viral load analysis with few patients. In contrast, a positive correlation between HPV DNA in blood with the severity or tumor size was observed in cervical [11,17,27,38–40], head and neck [41–43], lung [20], and breast cancer [44]. Thus, more studies are needed to clarify whether HPV DNA viral load can monitor cervical lesions.
Comment #5: In lines 154-156, the authors refer to Table 3. I cannot find Table 3, and I can't find the OR's they note. Was Table 3 deleted? Are the binomial logistic regressions they cite bivariate or multivariate?
Response: We deleted table 3. We performed the binomial logistic regression bivariate.
Comment #6: In Table 2, the p-values reported for the chi-square tests should be based on exact tests, rather than asymptotic tests. It is almost impossible to find a significant p-value for a chi-square when you have 1 or 2 counts in a cell.
Response: We mention in 152 lines: "*P-values were calculated by chi-square test or Fisher´s exact test. OR- Odds Ratio."
We used Fisher´s exact test for cells <5.

This manuscript is a resubmission of an earlier submission. The following is a list of the peer review reports and author responses from that submission.
Round 1
Reviewer 1 Report
In this manuscript the authors intend to characterise the correlation between HPV DNA in cervix/blood of patients with cervical lesions and healthy individuals. My overall comments are described in PDF comments of the manuscript.

Reviewer 2 Report
In the manuscript, Gomes do Nascimento et al. used paired peripheral blood and cervix samples obtained from 139 women, including healthy individuals, patients with CINI, CINII and CINIII lesions, and examined these for the presence of HPV DNA by PCR. The HPV positive cases were further genotyped.
With samples from 7 patients and 12 healthy women, which were HPV16 positive, the viral load was determined by semi-quantitative RT-PCR using the E6 RNA in paired cervix and blood samples. Furthermore, the physical state was analyzed by the amplification of the ORF for E2. The ratio E2/E6 was used to differentiate between integrated or mixed with mostly integrated or mostly episomal.
The authors found a high degree of HPV positivity in peripheral blood or cervix or both, the latter ranging from 24 to 44% of the cases. The authors demonstrate that the detection of HPV DNA in the cervix was associated with the presence of HPV DNA in the blood and the cervical lesions. Yet, there is no association between the detection of DNA in the blood and the cervical lesion.
The study presented in the manuscript may be a valuable contribution to answer the question, whether the analysis of the HPV DNA in peripheral blood may serve as a marker for screening for the presence of cervical lesions. Yet, the manuscript cannot be published in the present form. There are a number of corrections necessary prior publication, which are listed here:
- in Table 2: The formatting of table 2 should be corrected and adapted to one format. For instance, there is in one case a blank line between the parameter (Age and Residence) and not between the others. In addition, there is one lane displaced.
- The data in table 2 are not presented in a consistent manner. For example, in CINI/II or III section: total is missing, parentheses are missing (CINIII HPV pos it should be 5 (3,6%) ). In some sections (Residence, Number of pregnancies, Number of sexual partners, Previous STIs, and CINI/II or III) there are informations for some individuals missing. The authors should comment on this (for instance whether the informations are not available or what ever)
- to Figure 3 : the dots used to calculate the correlation shown in Fig. 3a and Fig. 3b are the same except one with obviously no viral load in the cervix. While in the b section (CINII/III patients) there are 6 dots/samples used for analysis and in the part (a) only comprising CINIII there are 7 dots/samples. This is not comprehensible. What about CINI lesions? In the Materials and Method section, it was stated that in addition to 7 paired samples from patient with CINI, II, and CINIII, 12 women with normal cytology were selected. What about these? Is there any correlation between viral load in cervix and blood in healthy individuals? The data used in Fig. 3a have to be checked and explained in more detail that the reader can follow the argumentation.
- I do not understand the sentence on p. 7, middle part: „It was also not observed integrated physical status in the blood samples from CIN patients.“ Please explain what is meant here.
- To Fig. 3 and 4: On the first view, the positive correlations seen in Fig 3a, b, and 4a, b are only due to the one patient which has obviously high viral loads and a mixed HPV DNA status in the blood as well as in the cervix. Are these Pearson correlation coefficients significant?
- The discussion should be shortened and focused more on the data and results presented in the manuscript.
Reviewer 3 Report
In this study, the authors have extracted DNA from matched cervical lesions and peripheral blood of women, including apparently healthy individuals and those with CINI-CINIII lesions. The DNA was submitted to HPV genotyping and for some samples estimates of the relative viral load and integration status also assessed. The presence in both specimen compartments were compared and correlated with clinical features and demographics.
Major Concerns:
- The underlying rationale for performing the study and contribution these studies will make to the understanding of the disease or in the current testing paradigm is not well defined, other than it has not been done.
- The specimen cohort selected is not adequate in size (not adequately powered), especially to support the multitude of clinical correlative analyses being performed. Rationale for such a small study size is warranted.
- It is unclear to this reviewer what the “cervix” samples studied were – formalin-fixed paraffin-embedded histology specimen or cytology specimen? This was very poorly detailed.
- The dataset is unclear: Does the “apparently healthy” group include those that were biopsy-proven negative? Many aspects of the dataset were poorly described.
- For the viral load assay, why were only random patients were selected for study? Was the standard curve run everytime the PCR was performed?
- The underlying data presented in Figure 1 should be provided as a supplementary table so that on a patient-by-patient basis the results can be examined.
- It is very surprising to this reviewer that the incidence of the high risk HPV is only found in about 50% of the cervical samples from patients with CINIII.
- In the discussion, it is unclear how the authors can state that only HPV DNA found in the cervix was associated with cervical lesions given the known presence of HR HPV that can clear.